# Optimized Multi-Epitope Norovirus Vaccines Induce Robust Humoral and Cellular Responses in Mice

**DOI:** 10.3390/vaccines14010050

**Published:** 2025-12-31

**Authors:** Ziyan Xing, Luyao Ji, Peifang Cao, Ercui Feng, Qing Xu, Xun Chen, Wenlong Dai, Nan Jiang

**Affiliations:** 1National Vaccine Innovation Platform, School of Pharmacy, Nanjing Medical University, Nanjing 211166, China; 2023110533@stu.njmu.edu.cn (Z.X.); jily199710@163.com (L.J.); 2023121601@stu.njmu.edu.cn (P.C.); chenxun@njmu.edu.cn (X.C.); 2The Affiliated Jiangning Hospital of Nanjing Medical University, Nanjing 211100, China; ercuifeng@163.com; 3Department of Infectious Diseases, The Affiliated Taizhou Peoples Hospital of Nanjing Medical University, Taizhou 225300, China; x_qing2@163.com; 4Taizhou School of Clinical Medicine, Nanjing Medical University, Taizhou 225300, China

**Keywords:** norovirus, multi-epitope vaccine, immunoinformatics, molecular dynamics, antibody response, T-cell immunity, vaccine design, reverse vaccinology

## Abstract

**Background**: Norovirus GII.4 is a major global health threat, yet no licensed vaccines exist due to the virus’s rapid evolution and high mutation rates. **Objective**: To rationally design and experimentally validate multi-epitope vaccine candidates against Norovirus GII.4 using computational immunoinformatics and in vivo evaluation. **Methods**: We employed reverse vaccinology to screen optimal norovirus GII.4 epitopes and systematically designed four construction strategies to evaluate different epitope topologies and adjuvants. Candidates underwent molecular dynamics simulations and were expressed in E. coli. Immunogenicity was assessed in BALB/c mice via ELISA and ELISPOT to evaluate humoral and cellular responses. **Results**: Three candidates (NV1, NV4, NV5) were successfully produced and induced cross-reactive antibodies against authentic GII.4 virus-like particles. Notably, the construction strategy influenced the immune response: NV5 (repetitive epitopes and HSP as adjuvant) elicited the highest antigen-specific antibody titers, NV1 (all types of epitopes and TLR as adjuvant) induced the strongest cellular response, and NV4 (repetitive epitopes and TLR as adjuvant) achieved the most rapid immune response. Consistently, in silico analysis showed that the NV1-TLR3 complex exhibits tighter interaction, higher binding energy, and greater structural stability, supporting its superior capacity to trigger cellular immunity. **Conclusions**: A rational multi-epitope vaccine design workflow successfully realized the translation from computational design to functional vaccines. Optimizing adjuvant selection and epitope construction is critical for eliciting immune responses in next-generation norovirus vaccines.

## 1. Introduction

Norovirus gastroenteritis is an acute intestinal infectious disease characterized primarily by sudden outbreak of diarrhea and/or vomiting, accompanied with nausea, abdominal pain, fever, and body aches [1]. Although viral gastroenteritis is typically self-limiting and resolves in one to three days, complications such as dehydration and shock cause severe consequences, particularly in vulnerable populations [2]. According to statistics from the World Health Organization (WHO) and the Centers for Disease Control and Prevention (CDC), norovirus causes over 684 million infections and more than 200,000 deaths annually worldwide [3,4].

Currently, there are no specific antiviral drugs or licensed vaccines for norovirus [5,6], making vaccine development a critical global health priority identified by WHO at the Product Development for Vaccines Advisory Committee meeting (PDVAC) in December 2023. Multiple norovirus vaccines are currently in clinical trials worldwide. Despite some high-profile failures, including HilleVax’s HIL-214/TAK-214 efficacy (5% in pediatric trials) (https://clinicaltrials.gov/search?id=NCT05281094, accessed on 15 December 2025 NCT05281094) [7,8], promising candidates have advanced to Phase III trials. Three vaccines are currently in Phase III: Moderna’s mRNA-1403 trivalent vaccine [9], enrolling 25,000 participants worldwide (currently on FDA hold) (https://clinicaltrials.gov/study/NCT06592794?id=NCT06592794&rank=1, accessed on 15 December 2025 NCT06592794) and Lanzhou Institute’s bivalent VLP vaccine (https://clinicaltrials.gov/study/NCT05916326?id=NCT05916326&rank=1, accessed on 15 December 2025 NCT05916326). Furthermore, Vaxart’s oral adenovirus vector vaccine demonstrates promising results with 85% viral shedding reduction in Phase II trials (https://clinicaltrials.gov/study/NCT05212168?id=NCT05212168&rank=1, accessed on 15 December 2025 NCT05212168) [10].

A fundamental challenge in norovirus vaccine development is to provide broad protection against genetically and antigenically diverse strains. Human infections are predominantly associated with the GI, GII, and GIV genogroups, with GII being the most prevalent [11,12,13]. This genotype exhibits exceptionally high mutation rates and rapid evolution, with new variants emerging every 2–4 years through immune evasion mechanisms [4]. Traditional vaccine approaches struggle to address it, necessitating innovative strategies for comprehensive strain coverage.

Multi-epitope vaccines represent a promising approach, comprising multiple T-cell and B-cell epitopes that elicit immune responses and immunological memory [14,15]. These peptide-based vaccines offer significant advantages, including cost-effectiveness, simplified manufacturing processes, enhanced stability, reduced toxicity, and population-specific optimization while avoiding adverse reactions associated with conventional vaccines [16]. Unlike traditional vaccines that may pose risks in immunocompromised populations due to infection-related mortality, diminished immunogenicity, and hypersensitivity reactions, multi-epitope vaccines provide safer alternatives with broad applicability across diverse conditions [17,18,19,20]. The hepatitis C antigenic peptide vaccine IC41 demonstrated excellent T-cell immune effects and low relapse rates in Phase I/II clinical trials [21], while multi-epitope vaccine strategies are being employed against drug-resistant microbes, *Mycobacterium tuberculosis*, and SARS-CoV-2 [22].

Previous computational efforts (e.g., Azim et al.; Ahmad et al.) proposed multi-epitope norovirus vaccines with favorable predicted properties and population coverage [23,24]. However, these studies remained at the in silico design stage without experimental validation. To overcome these two problems, four construction strategies were implemented in this work, and a multidimensional screening was performed by integrating physicochemical properties, structural stability with immune simulation and molecular dynamics, and subsequent validation of the vaccine efficacy via murine immunogenicity and cellular immunity assays. In addition, the reverse vaccinology-based computer-aided design was followed by comprehensive experiments to validate the murine immunogenicity and cellular immunity assays [25,26].

We selected the GII strain VP1 protein exhibiting the highest antigenicity for epitope prediction, integrating properties such as binding affinity, physicochemical characteristics, and allergenicity. Unlike traditional approaches involving simple linear epitope concatenation, we developed multiple diverse vaccine construction strategies, including varied topological positionings of epitopes, repetitive epitope patterns, and different adjuvant combinations. This systematic exploration investigated how the permutations of epitopes and adjuvants impact vaccine properties, spatial structure, and immune functions, utilizing computational methods and animal experiments.

This study successfully demonstrates the translation of reverse vaccinology approaches from theoretical design to functional immunogens. We hope a general framework for rational multi-epitope vaccine design methodology may be established.

## 2. Materials and Methods

The flowchart of designing a multi-epitope chimeric vaccine for norovirus is illustrated in Figure 1.

### 2.1. Screening for Optimal Antigenic Protein

A subtractive proteomics pipeline was employed to select the appropriate proteins for vaccine designing.

#### 2.1.1. Complete Retrieval of Norovirus Protein Sequences

The strains that cause human infection with norovirus mostly belong to the GI, GII, and GIV genogroups. So, their proteomes were retrieved from the National Center for Biotechnology Information (NCBI). To reduce the redundancy of proteins and enhance the efficiency of screening, homologous proteins were discarded by using the Cluster Data at High Identity with Tolerance (CD-Hit) server [27,28]. The threshold 60% was set to remove paralogous proteins showing >60% sequence similarity.

Then, to avoid the autoimmune response of the host, homologous proteins were identified by the BLASTp tool of the NCBI database from human (taxid: 9606) and three probiotics, including *Lactobacillus rhamnosus* (taxid: 47715), *Lactobacillus casei* (taxid: 1582), and *Lactobacillus johnsonii* (taxid: 33959). The criteria of sequence identity <30%, bit score of >100, and E-value threshold value of <0.005 were employed in this work [25,29,30,31].

#### 2.1.2. Transmembrane Helix Number Prediction

TMHMM [32] online servers were utilized to predict the number of transmembrane helices in the candidate proteins. Analyzing the transmembrane regions helps determine the subcellular localizations of proteins [33]. Proteins with transmembrane regions may function as membrane proteins, while proteins with fewer transmembrane helices may indicate roles as ex-proteome and secretome. The hydrophobic nature of membrane proteins, which typically renders them flexible and unstable, presents challenges in cloning, expression, and purification [34]. Proteins with fewer transmembrane helices often exhibit hydrophilic properties, making them excellent antigenic proteins, as the antigenic regions can be exposed for enhancing immunogenicity [35].

#### 2.1.3. Antigenicity and Allergenicity Prediction

The Vaxijen2.0 online server was employed to predict protein antigenicity [36,37]. The proteins with antigenicity scores > 0.5 were prioritarily selected, as this value indicates their strong potential to induce immune responses. Protein allergenicity was predicted by the AllerTop2.0 online server [38]. If the proteins were identified as allergenic, they were excluded from consideration.

### 2.2. Immune Cell Epitope Prediction

#### 2.2.1. Human Leukocyte Antigen (HLA) Restriction Determination

Before predicting immune cell epitopes, especially human T-cell epitopes, it is necessary to determine HLA restriction, which has implications for both identifying epitopes and population coverage of epitope-based subunit vaccines [39]. The specific HLA molecule responsible for binding and presenting the epitope is referred to as the HLA restriction [40]. It can be utilized to predict antigenic epitopes capable of binding to a specified demographic, thereby enhancing population coverage [41,42]. Certain epitopes can bind and be recognized by multiple HLA molecules and are referred to as “promiscuous epitopes”. Studies have shown that promiscuous epitopes are often immunodominant and significant for the immune response to a given antigen. Predicting such epitopes is crucial for the development of multi-epitope vaccines [43,44,45]. In other words, the predicted epitopes are based on hunting for the high-frequency HLA genotypes in a specific population, i.e., defining the HLA restrictions. We completed the task using the Allele Net Frequency Database.

#### 2.2.2. Cytotoxic T-Lymphocytes (CTL) Epitopes Prediction

A CTL epitope is a specific peptide that can be recognized by CD8+ T lymphocytes and bind to MHC-I molecules on their surface [46,47,48,49]. In the process of T-lymphocytes epitope prediction, using the specific HLA genotypes determined in the previous step as input parameters, we employed several online servers to collectively predict the most accurate epitopes. We collectively employed the IEDB-NetMHCPan 4.1 EL [50,51,52,53], NetCTL [54], IEDB-Consensus [55,56], and MHCPred [57,58] web servers to predict CTL epitopes. All URLs have been provided in the Appendix B for accessibility.

#### 2.2.3. Helper T-Lymphocytes (HTL) Epitope Prediction

An HTL epitope is a specific peptide that can be presented by MHC-II molecules and recognized by CD4+ T lymphocytes. Primed CD4+ T-cells become helper T-cells, which can amplify the immune response [59]. Similar to CTL epitope prediction, the same algorithms and web servers were utilized to predict HTL epitopes [60,61,62,63].

#### 2.2.4. Linear B-Lymphocytes (LBL) Epitope Prediction

A B-cell epitope is the solvent-exposed portion of the antigen, binding to the immunoglobulin or antibody, that elicits a cellular or humoral immune response [59,64]. We used the online servers ABCpred [65] and BepiPred-2.0 [59,66] to predict LBL epitopes.

#### 2.2.5. Population Coverage and Various Property Analysis of Predicted Epitopes

It is essential to conduct population coverage analysis of the HLA alleles corresponding to the predicted epitopes to assess the hit rate of the epitopes and anticipated immune response in the target population [67,68]. It is preferable to select epitopes that combine with multiple alleles, the “promiscuous epitopes” described in Section 2.2.1, which have high population coverage.

To be able to safely use the predicted epitopes in vaccine construction, we also need to analyze their allergenicity and toxicity, respectively, in the AllerTop2.0 [38,69] and ToxinPred servers [70,71].

In addition, the hydropathicity of epitopes can be analyzed on the hydropathicity server [72]. The epitopes were chosen to be more hydrophilic to be exposed on the surface of the chimeric vaccine [69,70]. They will be more likely to be recognized and bound by the immune cells for an immune response.

### 2.3. Construction and Screening of Multi-Epitope Vaccines

#### 2.3.1. Construction of Vaccine Molecules

The subunit vaccines offer stability, safety, and ease of production, but their intrinsic immunogenicity is modest. So adjuvants are essential to enhance immune response and immune memory [73,74]. Guided by norovirus biology, we prioritized endosomal TLR7/TLR3 agonists to trigger the IRF3/IRF7 signaling pathway and drive antiviral Th1/IFN-γ/CTL responses [75,76]. In addition, we appended PADRE (pan-DR helper epitope) to provide broad HLA-DR coverage and stable, universal CD4+ T-cell help [77]. Furthermore, heat shocked protein (HSP) also can be used as an adjuvant to facilitate antigen cross-presentation in dendritic cells, thereby amplifying downstream signaling pathways essential for CD8+ cytotoxic T lymphocyte responses [78]. Within this framework, the multi-epitope vaccine sequence comprised adjuvants, a polyhistidine tag (His-tag), CTL epitopes, HTL epitopes, and LBL epitopes, connected by purpose-chosen linkers: EAAAK (adjuvant spacings), RVRR (His-tag junction), AAY (CTL epitopes spacing), GPGPG (HTL spacing), and KK (LBL spacing) [79].

In this study, we pre-specified four construction strategies for multi-epitope vaccine construction (as depicted in Figure 2) to test different immunological hypotheses while keeping linkers and epitope sources constant. Strategy I (all predicted epitopes + TLR7/TLR3 agonist + PADRE at termini) was intended to maximize breadth while testing whether epitope or adjuvant topology alters predicted antigenicity and physicochemical properties (Figure 2a). Strategy II (one optimal epitope for CTL/HTL/LBL + adjuvants) was designed to reduce sequence length and improve expression/solubility (Figure 2b). Strategy III (six times repetition of one epitope) explicitly tested whether it could enhance the humoral immunity and immunogenicity of vaccines, as well as reduce mutations in subsequent plasmid expression [80,81] (Figure 2c). Strategy IV (replacing TLR agonists with HSP65 and HSP70_407–426_) evaluated whether HSP could enhance cellular immune responses through cross-presentation and improve physicochemical stability, serving as a safe and well-tolerated alternative to synthetic TLR agonists (Figure 2d). These construction strategies covered breadth, topological position, and adjuvant class, thereby enabling a comprehensive interpretation of the relationships among sequence, structure, and immune function.

#### 2.3.2. Antigenicity, Allergenicity, and Physical/Chemical Properties of Vaccine Constructs

Before any structural modeling or animal studies, candidates were required to be non-allergenic as predicted by AllerTOP, have antigenicity > 0.5 by VaxiJen, solubility > 0.45 as assessed by Protein-Sol [82], instability index < 40, a favorable aliphatic index with negative GRAVY as computed in ProtParam, and a sufficiently long predicted half-life as estimated by ProtParam [83].

### 2.4. Vaccine Immune Simulation

We used the online immune simulation server C-IMMSIM to simulate the immune responses of vaccine candidates at days 0, 28, and 56 post vaccination within 365 days [84,85,86]. In the server, each time step is set to 8 h, resulting in a total of 1095 steps. The host HLA typing selected the relatively common frequencies of A02:01, A24:02, B56:01, B07:02, DRB115:01, and DRB103:01. C-ImmSim is an agent-based immune response prediction model that uses position-specific scoring matrices (PSSM) to simulate the immune system’s response to pathogens. It utilizes a three-dimensional cellular automaton to represent immune cells and simulates their interactions under specific rules. The model represents molecules in a lattice manner and simulates the bone marrow, thymus, and lymphoid organs [87].

### 2.5. Prediction and Validation of Vaccine Structure

#### 2.5.1. Secondary and Tertiary Structure Prediction

We employed the PSIPRED4.0 server for predicting the secondary structures of our candidate vaccines [88,89], followed by the Robetta online server to determine their tertiary structures with high fidelity [90,91]. This process helps us anticipate their stability, potential biochemical functions and roles [92,93].

#### 2.5.2. Validation of Structural Conformation Rationality and Refinement

On the Robetta online server, each vaccine sequence is predicted to have 5 models, and we can select the most suitable model for each sequence by analyzing their structural rationality. ProSA-web can evaluate protein structures by identifying errors and comparing models to native proteins, ensuring structural accuracy and reliability. Additionally, the Ramachandran plot, as analyzed by SAVES v6.0, visually assesses protein model quality by mapping the phi and psi torsion angles, indicating the plausibility of amino acid residue conformations within the protein backbone [94,95]. Subsequently, structural refinement was performed using the FG-MD server [96].

### 2.6. Molecular Docking with TLR3 and Interaction Analysis

We utilized the online server ClusPro2.0, a protein–protein docking tool, to dock the remaining candidate vaccines individually with TLR3(PDB ID: 7C76) [97,98,99,100]. This was performed to assess the docking quality and the degree of interaction between them. TLR3 is a cellular recognition system capable of identifying viral nucleic acids, making it a suitable target for norovirus antigen docking.

Building upon the docking results, we conducted further analysis of the docking model interactions using PDBePISA [101,102]. Additionally, we utilized LigPlus software to generate an interaction diagram illustrating the interactions [103,104].

### 2.7. Molecular Dynamics (MD) Simulation

We utilized Amber [105,106,107] to perform MD simulations on four vaccine-TLR3 complexes to examine whether their molecular structures could remain stable during 150 ns. The complexes were placed in a TIP3P water box and supplemented with Na^+^ and Cl ^−^ ions to mimic a physiological saline environment (0.9% NaCl). Stability was verified through root mean square deviation (RMSD). In addition, we performed the /PBSA method to calculate binding energy of vaccine candidates for TLR3.

### 2.8. Codon Optimization and In Silico Cloning

Finally, we performed codon optimization to align gene sequences with the *Escherichia coli* expression system and improve protein expression efficiency [108,109]. We targeted a GC content within the range of 45 to 60% to ensure the structural stability of the gene sequences and aimed for a Codon Adaptation Index (CAI) exceeding 0.7 to enhance their translational efficiency in the host organism [110,111,112]. Following this, we utilized the SnapGene tool to insert the codon-optimized sequences into the pET-28a (+) *E. coli* plasmid vector, preparing for future protein cloning and expression.

### 2.9. Expression and Immunological Evaluation of Vaccine Candidates

#### 2.9.1. Expression and Purification

Optimized gene sequences synthesized by TsingKe Biotechnology were inserted into the pET28a vector using the BamHI and SacI restriction sites. Plasmids were transformed into BL21 *E. coli* competent cells via chemical transformation. Transformed BL21 cultures were grown at 37 °C, 220 rpm shaking until $OD_{600}\approx 0.5$∼0.8. Protein expression was induced with 0.1 mM IPTG (final concentration). Cultures were shifted to 16 °C and shaken at 220 rpm for 16–18 h (overnight). Cells were harvested by centrifugation at 5000 rpm for 5 min. Cell pellets were resuspended in lysis buffer (20 mM Tris-HCl, pH 8.0; 500 mM NaCl). Cells were lysed using a high-pressure homogenizer at 800 MPa for 3 min. Lysates were clarified by centrifugation at 12,000 rpm for 10 min. For soluble protein: The supernatant was collected, and the soluble protein was purified via Ni-affinity chromatography. For inclusion body protein: The pellet (inclusion bodies) was collected, resuspended in denaturing buffer (6 M urea, 20 mM Tris-HCl, pH 8.0, 500 mM NaCl), and centrifuged again at 12,000 rpm for 10 min. The supernatant containing solubilized inclusion body proteins was then purified via Ni-affinity chromatography.

#### 2.9.2. Transmission Electron Microscopy (TEM) 

Purified GII.4 virus-like particles (VLPs) were generously provided by Professor Zhong Huang from Fudan University [113], and samples were visualized using transmission electron microscopy (TEM).

#### 2.9.3. Mouse Immunization

Antigens were emulsified with Freund’s adjuvant according to the manufacturer’s instructions. Groups of six BALB/c mice received subcutaneous injections on day 0, 21, and 35. Injections contained the following: PBS (control), candidates emulsified in adjuvant. Freund’s complete adjuvant (FCA) was used for the prime immunization (day 0). Freund’s incomplete adjuvant (FIA) was used for booster immunizations (day 21 and 35). Blood samples were collected on day 21, 35, and 49 post-prime immunization. Mice were euthanized 14 days after the final immunization, and spleens were harvested for interferon-gamma (IFN-γ) secretion assessment.

#### 2.9.4. Exploring the Specificity of Antibody Responses with Vaccines

Serum levels of antigen-specific antibodies were measured by indirect ELISA, adapted from a previously described method [113], with the following modifications: 96-well plates were coated with either 30 ng of purified candidates or 10 ng of norovirus GII.4 VLP per well. To determine endpoint titers, all serum samples were tested using serial dilutions. The endpoint titer was defined as the highest dilution yielding a positive result. Sera from the PBS control group remained negative at the lowest dilution tested (1:64) and were assigned a titer of 32 for geometric mean titer (GMT) calculation.

#### 2.9.5. ELISPOT Assay

Spleens were aseptically dissociated by gentle grinding through a cell strainer using a sterile syringe plunger. Cells were treated with 1× RBC lysis buffer to remove erythrocytes. After RBC lysis, cells were washed and resuspended in RPMI medium supplemented with 10% FBS to a concentration of $1.0\times 10^{7}$ cells/mL. $1.0\times 10^{5}$ cells were plated per well and incubated overnight in a cell culture incubator. Cells were stimulated with purified norovirus GII.4 VLP for 24 h. IFN-γ secretion was detected using a Hunan IFN-γ precoated ELISPOT kit (Dakewe) according to the manufacturer’s protocol. Spot-forming units (SFUs) were enumerated automatically using an ELISPOT plate reader.

#### 2.9.6. Statistical Analysis

All statistical analyses were performed using GraphPad Prism software (version 8.0). All data were expressed as the mean ± SD. Differences between groups were examined for statistical significance using a one-way analysis of variance (ANOVA) with Tukey’s multiple comparison post-test.

#### 2.9.7. Ethical Statement

The use of animals was approved by the Institutional Animal Care and Use Committee of Nanjing Medical University (IACUC-2411051) and followed the relevant regulations of national animal welfare ethics.

## 3. Results

### 3.1. Identification of GII VP1 as Optimal Antigen with High Antigenicity and Safety

According to the results obtained from NCBI, each strain of norovirus GI, GII, and GIV has three types of proteins, containing non-structural polyprotein, capsid protein VP1, and capsid protein VP2. Therefore, a total of nine protein sequences were retrieved. One redundant protein was removed on the CD-HIT server. These virus proteins had no homology with host proteins, *Lactobacillus casei* and *Lactobacillus johnsonii*, according to the Blastp homology identification analysis. However, when compared with the complete set of proteins from *Lactobacillus lactis*, two homologous protein sequences, GI and GIV polyproteins, were found and removed from the candidate protein sequences (Appendix A). The remaining six candidate protein sequences had a predicted transmembrane helix count of 0 according to TMHMM and HMMTOP predictions, meeting the criteria of having 0 or 1 transmembrane helices, which facilitates ease of protein expression and purification. Subsequently, GII VP2 and GIV VP2 were excluded because of allergenicity (Appendix A). Among the remaining four candidate proteins, the protein GII VP1, with the highest antigenicity score of 0.55, was selected as the optimal antigenic protein.

### 3.2. Screening and Selection of Immune Cell Epitopes

In this work, we first surveyed the most frequent HLA genotypes in the population using the allele frequency database [114]. Among MHC-I molecules, HLA-A*02:01, HLA-A*24:02, HLA-A*11:01, HLA-C*04:01, HLA-C*03:04, and HLA-B*56:01 are relatively common. For MHC-II molecules, HLA-DRB1*15:01, HLA-DRB1*03:01, HLA-DQA1*05:01, and HLA-DQA1*03:01 appear frequently. These HLA restrictions are used to predict CTL epitopes and HTL epitopes. Additionally, 9 CTL epitopes (Appendix A), 7 HTL epitopes (Appendix A), and 9 linear LBL epitopes (Appendix A) were predicted. The percentile rank of predicted CTL and HTL epitopes was also explored, where lower values indicate better binding affinity with MHC molecules. AllerTop2.0 and ToxinPred were used to analyze allergenicity and toxicity, with the results listed in the tables. The preliminarily screened epitopes meeting the criteria of being non-allergenic, non-toxic, highly antigenic, top-ranked by percentile, and capable of binding to restrictive HLA supertypes were compiled in Table 1, totaling 5 CTL epitopes, 2 HTL epitopes, and 2 LBL epitopes. Sequence alignment within Discovery Studio software showed that these epitopes also contain conserved regions of the VP1 sequence among epidemic strains, and the linear B-cell epitopes were all located within the P structural domain. Additionally, considering high population coverage and high hydrophilicity are desirable characteristics of antigenic epitopes, these values were also predicted by the lEDB Population Coverage and ToxinPred tools, respectively.

### 3.3. Four Construction Strategies and Screened Vaccine Candidates

Guided by these four construction strategies, we generated 480, 12, 6, and 3 permutations under Strategies I, II, III, and IV, respectively (Figure 2). Their physicochemical profiling, including antigenicity, solubility, predicted half-life, and stability metrics (instability index, aliphatic index, GRAVY), were shown in Appendix A. Constructs with VaxiJen < 0.5 were removed. Within Strategy I, the 480 permutations showed highly similar profiles; we therefore advanced a single representative (7PC12345H12B123) for downstream analyses. Candidates with short predicted half-lives were excluded (e.g., 3PCHB7 and 3PCCCCC7: 1.1 h in mammalian reticulocytes; 2 min in yeast and 2 min in *E. coli*). We required Protein-Sol > 0.45; sequences below this threshold were discarded. Remaining constructs met instability index < 40, showed favorable aliphatic indices, and had negative GRAVY values. In total, six vaccine sequences (NV1–NV6, with full names given in parentheses) satisfied all filters and are listed in Table 2.

### 3.4. Immune Simulation Predicts Robust Immune Responses for Vaccine Candidates

We used the C-IMMSIM server to simulate the immune response of remaining vaccine candidates and utilized various antibodies, cytokines, and immune cell levels changes to characterize immune response effects. It is evident from Appendix A that the vaccination of vaccine molecules at three time points of 0, 28, and 56 days could cause an increase in immunoglobulin (antibody) levels, and all antibody indicators were predicted to be activated. Furthermore, compared to the primary reaction, the increase in the second and third order reactions also was predicted to be more significant. These modeling results were in good agreement with the ideal immune response trend. In addition, vaccination results revealed vaccination could trigger high levels of IFN-γ and IL-2, both of which are important to regulate and promote immune responses [115,116].

Analyzing the immune simulation results of the six vaccine candidates revealed that modeling predicted NV6 could elicit remarkably high antibody titers with an appropriate ratio of IgM to IgG levels, along with predicted significant increases in other immune factors and cell levels (Appendix A). Among the single-epitope constructed sequences, NV1 modeling revealed it could produce high antibody titers, and the levels of other immune cells could be satisfactory (Appendix A). In comparison, the antibody titers for NV2 and NV3 modeling were predicted to be lower, and the helper T-cell counts post-secondary vaccination were not predicted to exhibit a better response compared to the primary vaccination. Additionally, the macrophage population was predicted to undergo apoptosis prematurely (Appendix A). Therefore, candidates NV2 and NV3 were excluded based on the modeling results. As for NV4 modeling, its antibody titer was predicted to be very high. However, both in the early and later stages, the predicted content of IgM was always higher than that of IgG (Appendix A). This suggested that NV4 could have the potential to combat acute norovirus infection. Therefore, it was preserved. The predicted immune simulation results of NV5 modeling were not as good as the previous models (Appendix A). However, considering its excellent physicochemical properties, such as high aliphatic coefficient, strong protein solubility, moderate hydrophobicity, etc., it remained for subsequent analysis and research.

### 3.5. Predicted Secondary/Tertiary Structure of Vaccine Candidates

The secondary and the optimized tertiary structures refined of the remaining four vaccine candidates were modeled as shown in Figure 3. Using the Robetta server, five tertiary structure models with the highest scores were predicted for each vaccine candidate. We selected the most appropriate model for each sequence by analyzing their conformational rationality, which was determined by smaller local energy fluctuations, fewer errors in chemical conformation analysis, and fewer amino acids in the disallowed regions of the Ramachandran plot. These criteria were supported by data obtained from the ProSA-web server and the SAVES 6.0 server, which can assess the conformational validity of the tertiary structures.

The tertiary structure validations for the four candidates are all presented in Figure 4. We used Z-score on the ProSA-web server to evaluate the overall model quality, which can be used to check whether the Z-score of the input structure is within the range of scores typically found for native proteins of similar size. Local model quality was characterized by an energy function of amino acid sequence. The above results collectively indicated that the predicted models were rational and reliable when assessed for both overall and local quality.

The Ramachandran plot results analyzed by the SAVES v6.0 server are shown in Figure 4. The Ramachandran plot delineates the permissible regions for the phi (ϕ) and psi (ψ) torsional angles within the protein’s backbone. It reflects whether the conformations of individual amino acid residues within the protein structure are plausible. If more than 90% of amino acids fall within the allowed region, the stereochemistry is considered acceptable, and fewer amino acids falling within the disallowed region are preferable. The results showed that the vast majority of amino acids in the candidates fall within the allowed regions, with only 0.8–3.3% of amino acids residing in the disallowed regions.

### 3.6. Theoretical Analysis of Interactions Between Vaccine Candidates with TLR3

The docking results between vaccine candidates and TLR3 are shown in Figure 5. It is evident that the candidates NV6, NV1, and NV4 interact with TLR3 tightly and deeply, while the interacting region of candidate NV5 is quite limited and appears unstable. Furthermore, we deeply analyzed the interactions of the docking models using PDBePISA (Table 3), and the interaction residues were illustrated using LigPlot+ software (Appendix A). From the results shown in the table, NV1 and NV4 complexes maintained a higher proportion of interface and a greater number of interactions (including hydrogen bonds and salt bridges). In contrast, NV5 and NV6 complexes displayed lower interfacial proportion and fewer interactions.

Furthermore, to examine the structural stability of the constructed vaccine candidates in vivo solvent environment, we conducted a 150 ns molecular dynamics simulation. As shown in Figure 6, the RMSD curves of NV1, NV4, and NV6 docking complexes tended to stabilize and indicated that their structures remained relatively stable throughout the MD simulation. In contrast, the NV5 docking complex showed substantial RMSD fluctuations, implying structural instability in its interaction with TLR3. In addition, based on the simulation trajectories, we calculated the binding free energies of each vaccine for TLR3 using the MM/PBSA method (Appendix A). The results revealed that NV1 exhibited the highest binding energy, followed by NV4, whereas NV5 and NV6 showed relatively lower values. These findings are consistent with the interaction analysis and RMSD results discussed above.

### 3.7. Codon Optimization and In Silico Cloning into pET-28a(+)

Codon optimization adjusts gene sequences to match host species preferences, enhancing protein expression and synthesis efficiency by reducing rare codon usage and aligning with abundant tRNAs. GC content refers to the percentage of guanine and cytosine bases in the gene sequence. In this study, we chose the *Escherichia coli* expression system, and accordingly, the GC content is recommended to be 45–60%. CAI refers to the degree of similarity between the codon usage frequency of the heterologous mRNA sequence and the optimal codon usage frequency in the host cell. Generally, a CAI above 0.7 indicates good protein expression efficiency.

The remaining candidates achieved good levels of CAI and GC content after codon optimization on Jcat, as shown in Appendix A. Based on these results, the designed vaccine sequences were then expressed into the pET-28a (+) vector through SnapGene, as represented in Appendix A.

### 3.8. Expression and Purification of Target Antigens for Immunological Assays

Gene synthesis assessment indicated that NV6 (7PC12345H12B123) has 2 transmembrane helices, which hinder protein expression and purification (Appendix A); thus, we proceeded with the synthesis of the following three candidates: NV5 (65PCCCCCCM2): Comprises 6×CTL epitopes of norovirus GII.4, linked by one HSP65, one PADRE, and two HSP70 molecules. A 6×His tag was added to the C-terminus. NV1 (7PCHB3): Comprises 6×CTL epitopes linked by one TLR7 agonist, one PADRE, and one TLR3 agonist. A 6×His tag was added to the C-terminus NV4 (7PCCCCCC3): Comprises one CTL epitope, one HTL epitope, and one LBL epitope linked by one TLR7 agonist, one PADRE, and one TLR3 agonist. A 6×His tag was added to the C-terminus (Figure 7A).

SDS-PAGE analysis confirmed the purity and size of the expressed antigens: Purified NV5 displayed a dominant band at ∼76 kDa, and these observed molecular weights were consistent with predicted sizes (Figure 7B). Purified NV4 displayed a dominant band at ∼35 kDa, and Purified NV1 displayed a dominant band at ∼35 kDa (Figure 7B); this is slightly larger than our predicted molecular weight (NV4 ∼31 kDa, NV1 ∼27 kDa). These proteins are unusually basic (NV4: K/R/H ≈ 19.4 of residues, NV1: K/R/H ≈ 23% of residues) and rich proline (NV4: P ≈ 6.5% of residues, NV1: P ≈ 4.8% of residues). These features can reduce uniform SDS binding and result in slower migration on SDS–PAGE [117,118], producing an apparently higher apparent molecular weight.

### 3.9. Specificity of Antibody Responses to Vaccines

To evaluate immunogenicity, groups of mice were immunized subcutaneously with NV5, NV1, or NV4 on days 0, 21, and 35; a control group received PBS. Serum samples were collected on days 21, 35, and 49 post-primary immunization (Figure 7C). The antigen-specific geometric mean titers (GMTs) against the respective immunizing antigens were determined as shown in Figure 7D. No significant difference in antigen-binding GMT was observed between NV1 and NV4 sera, indicating comparable ability to induce antigen-specific antibodies. Sera from NV5-immunized mice exhibited significantly higher GMTs than those from NV1 and NV4 groups, demonstrating NV5’s superior capacity to induce antigen-specific antibodies. None of the groups showed boosting after the 2nd dose (Appendix A). Similar observations have been reported for norovirus vaccine immunogens, where the highest antibody titers were achieved rapidly after the first or second dose, with limited further increase upon additional boosting [119,120].

### 3.10. Norovirus GII.4-Specific Immune Responses Elicited by Vaccines

Prior to assessing GII.4-VLP binding titers, purified norovirus GII.4 VLPs were characterized by transmission electron microscopy (TEM), confirming their spherical morphology with a diameter of ∼38 nm (Figure 8A). GII.4-specific geometric mean titers (GMTs) were determined using serially diluted serum samples: Post-prime immunization: 1109 for NV5-immunized serum, 683 for NV1-immunized serum, 1536 for NV4-immunized serum. Post-second immunization: 1365 for NV5-immunized serum, 1877 for NV1-immunized serum, 1877 for NV4-immunized serum. Post-third immunization: 1707 for NV5-immunized serum, 1109 for NV1-immunized serum, 4352 for NV4-immunized serum (Figure 8B). Only NV4 sera showed significantly higher GII.4-VLP binding GMTs than NV5 and NV1 after the prime immunization. No significant differences in GII.4-VLP-specific antibody titers were observed between the three vaccine groups after boost immunizations, indicating that all three vaccines possess comparable ability to induce GII.4-specific antibodies. However, the NV4 vaccine induced GII.4-specific antibodies more rapidly.

To evaluate GII.4-specific interferon-gamma (IFN-γ) production, splenocytes were isolated 14 days after the final immunization, stimulated ex vivo with GII.4-VLP, and assessed using ELISPOT (Figure 8C). ELISPOT analysis revealed that all three vaccines effectively stimulated GII.4-specific IFN-γ secretion. Notably, splenocytes from NV1-immunized mice produced significantly more IFN-γ spots than those from NV5 and NV4 groups (Figure 8D), suggesting that the NV1 vaccine may elicit a stronger cellular immune response. This is consistent with our molecular docking results: the NV1–TLR3 complex had higher stability than the NV4–TLR3 and NV5–TLR3 complexes, and moreover, the NV1–TLR3 complex shows higher binding energy and a greater number of interfacial interactions.

## 4. Discussion

Norovirus remains a major global health threat, with no licensed vaccines or specific antivirals currently available. The rapid evolution and high mutation rate of GII.4 strains, which cause 60–90% of global outbreaks, challenge traditional vaccine development due to antigenic drift every 2–4 years.

A key achievement was translating computationally designed vaccines into functional immunogens. The three most promising candidates—NV5(65PCCCCCCM2), NV1(7PCHB3), and NV4(7PCCCCCC3)—were expressed in *E. coli*, purified by Ni-affinity chromatography, and confirmed by SDS-PAGE at predicted molecular weights (76, 27, and 31 kDa). Importantly, animal immunization further confirmed their ability to elicit specific immune responses, supporting their potential as vaccine candidates.

Mouse immunizations provided critical validation of computational predictions. Experimental immunogenic profiles aligned closely with simulations and immunoinformatics analyses. The NV5 construct, carrying six repetitive CTL epitopes with HSP65/70 adjuvants, induced exceptionally high antibody titers (GMT 251,221 after the second dose), attributable to multivalent epitope presentation and enhanced B-cell activation. Specifically, NV5 elicits markedly high GMTs, likely reflecting the strong immunostimulatory effects of HSP65 and HSP70; however, early VLP-specific responses are more pronounced in NV4. This is attributable to the presence of TLR7 and TLR3 in NV1 and NV4, which facilitate rapid VLP-specific activation, while the multiple tandem CTL epitopes in NV4 further accelerate the induction of specific antibodies. At later time points, antibody titers converge across all groups, consistent with the shared TLR sequence incorporated in each construct. In brief, NV1 contains TLR7 and TLR3 agonists, both of which promote a strong Th1-biased response by inducing type I interferons, IL-12, and robust dendritic cell activation. TLR7 stimulation in plasmacytoid DCs drives IFN-α/β production, whereas TLR3 activation triggers TRIF-dependent signaling and high IL-12 levels. These cytokines collectively enhance CD4+ Th1 differentiation and promote IFN-γ production, which is consistent with the strong IFN-γ responses observed in Figure 8D. Furthermore, the strong cellular immunity elicited by NV1 aligned well with the results of in silico simulation. Molecular docking and MD simulation demonstrated that the NV1-TLR3 complex exhibits tighter interaction, higher binding energy, and superior structural stability.

A significant finding was that all three vaccines induced cross-reactive antibodies recognizing authentic GII.4 VLPs. Given GII.4’s dominance and rapid evolution, this cross-reactivity is crucial. Notably, NV4 induced GII.4-specific antibodies most rapidly, with detectable responses after the first immunization and peak titers (4352) after the third, suggesting advantages for rapid outbreak response. By the end of boosting, comparable VLP binding titers were achieved across candidates, validating our epitope selection targeting conserved VP1 regions likely preserved across GII.4 variants and other genotypes.

Our results highlight how vaccine construction influences immune outcomes. NV5’s superior capacity to induce antigen-specific antibodies and the high antibody titers observed for NV4 during primary immunization confirm that repetitive epitope designs favor humoral immunity. In contrast, NV1 elicited stronger cellular immune responses than other strategies, highlighting the influence of diverse epitope composition and adjuvant selection in driving robust cellular immunity. It is a pity that our results show that the antibodies cannot prevent VLP interaction with HBGAs. This is a limitation of our study and may be due to the fact that the designed vaccine includes only one binding epitope of GII.4, resulting in antibodies that are unable to effectively prevent the VLP interaction with HBGAs.

NV6 designed by strategy I (all nine epitopes + TLR3/7 agonists + PADRE) achieved broad coverage but low antigenicity, likely due to epitope dilution and suboptimal adjuvant ratios (Figure 7A). NV1 designed by strategy II (optimized epitope selection with the same adjuvants) enhanced the structural stability of the docking complex and promoted tighter interactions, and the resulting constructs also elicited the strongest cellular immune responses among all strategies. NV4 designed by strategy III (repetitive epitopes) maximized antibody induction and validated predicted IgM predominance, supporting use in acute infection control. NV5 designed by strategy IV (HSP as adjuvant) significantly enhanced physicochemical stability. Interestingly, despite exhibiting structural instability and limited interactions in the vaccine–TLR3 docking complex, this strategy showed a superior capacity to induce antigen-specific antibodies.

Several limitations should be noted. First, results are limited to murine models, and translation to humans must consider species-specific immunity and HLA diversity. Second, the resulting antibodies showed GII.4 VLP interaction but barely prevent VLP binding with HBGAs. IgA was not detected in the serum.

## 5. Conclusions

This study theoretically designed and experimentally validated multi-epitope subunit vaccines against norovirus using reverse vaccinology and computational approaches. Three vaccine candidates, NV5 (65PCCCCCCM2), NV1 (7PCHB3), and NV4 (7PCCCCCC3), were successfully expressed, purified, and demonstrated distinct immunological properties in murine models. All candidates induced cross-reactive antibodies against GII.4 virus-like particles, with 65PCCCCCCM2 showing superior capacity to induce antigen-specific antibodies, 7PCHB3 eliciting the strongest cellular immunity, and 7PCCCCCC3 providing rapid immune priming. Our comparative analysis of four construction strategies revealed that vaccine constructions significantly influence immune outcomes: repetitive designs favored antibody responses, optimized epitope selection enhanced adjuvant effects, and balanced architectures achieved comprehensive immunity. The successful translation from computational design to functional vaccines demonstrated the potential of rationally designed multi-epitope vaccines for norovirus prevention and established a validated framework for developing vaccines against rapidly evolving viral pathogens. Future work will focus on designing vaccines that incorporate multiple epitopes from different norovirus serotypes, so that we can generate broad neutralizing antibodies.

## Figures and Tables

**Figure 1 vaccines-14-00050-f001:**
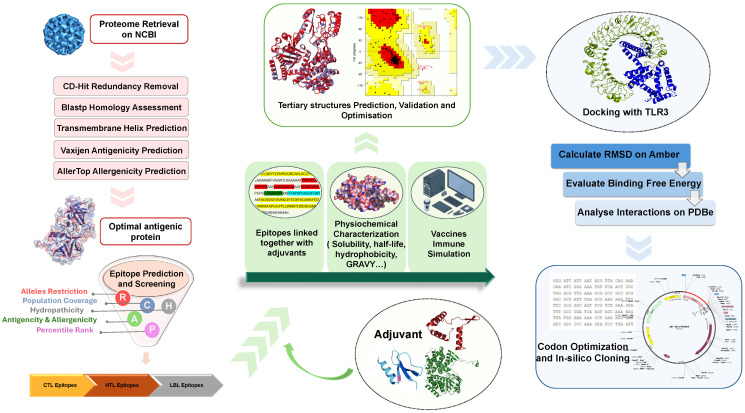
Flowchart for designing a multi-epitope vaccine of norovirus based on immunoinformatics. The asterisk is an automated annotation generated by the molecular biology software SnapGene. It indicates methylation sensitivity.

**Figure 2 vaccines-14-00050-f002:**
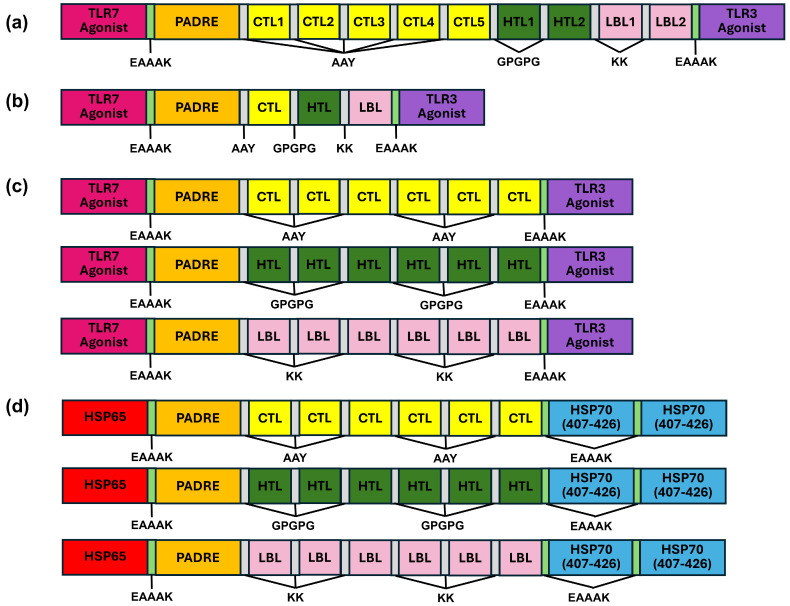
Schematic diagrams of multi-epitope vaccines. (**a**) Strategy I: vaccines with all predicted epitopes plus TLR7, TLR3 agonists, and PADRE. (**b**) Strategy II: vaccines with selected epitopes plus TLR7, TLR3 agonists, and PADRE. (**c**) Strategy III: repeated-epitope vaccines (six repeats) with TLR7, TLR3 agonists, and PADRE. (**d**) Strategy IV: repetitive-epitope vaccine (six repeats) with HSP65, HSP70 (407–426), and PADRE.

**Figure 3 vaccines-14-00050-f003:**
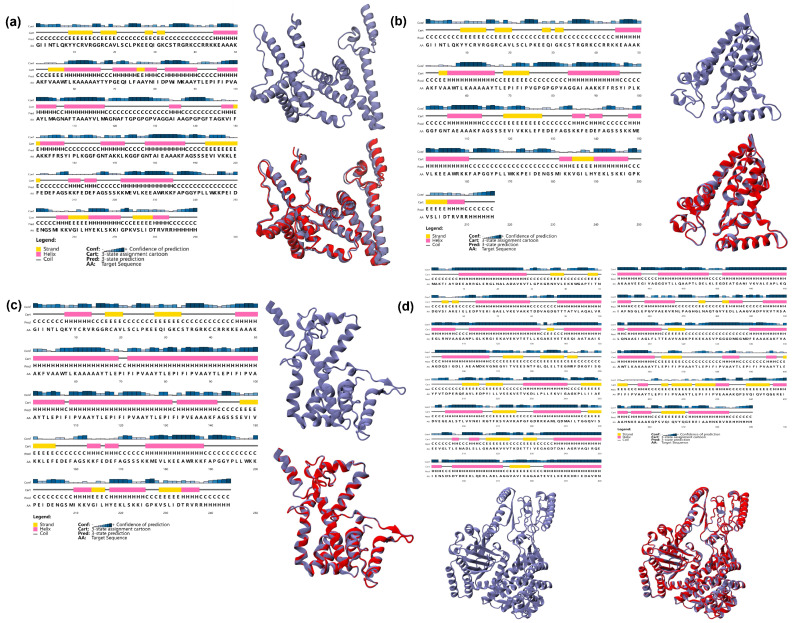
The secondary and tertiary structures of vaccine candidates of (**a**) NV6, (**b**) NV1, (**c**) NV4, and (**d**) NV5. The red-colored models are refined tertiary structures.

**Figure 4 vaccines-14-00050-f004:**
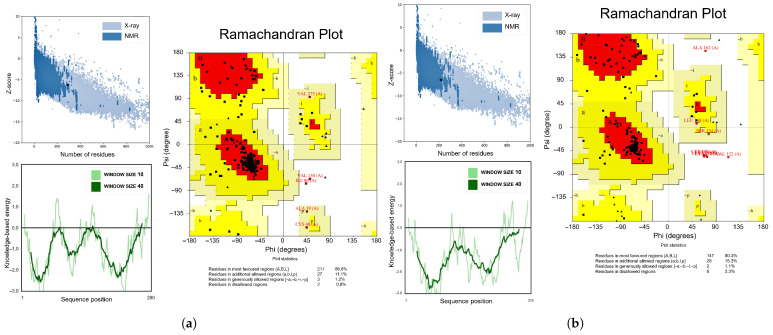
Comprehensive validation of the tertiary structures for four vaccine candidates of (**a**) 7PC12345H12B123, (**b**) NV1, (**c**) NV4, and (**d**) NV5.

**Figure 5 vaccines-14-00050-f005:**
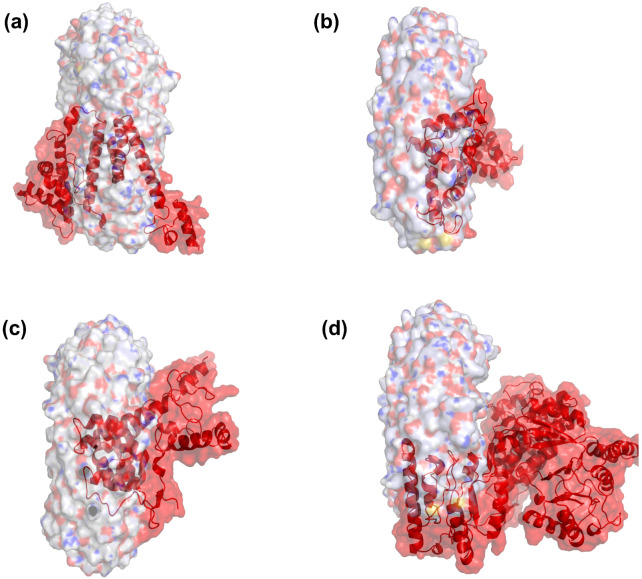
Side views of docked complexes of human Toll-like receptor 3 (TLR3, shown in solid red) with (**a**) NV6, (**b**) NV1, (**c**) NV4, (**d**) NV5.

**Figure 6 vaccines-14-00050-f006:**
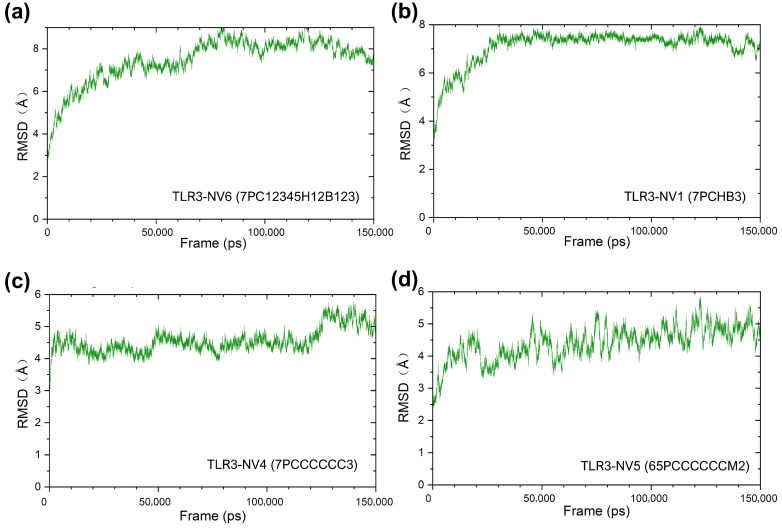
Root mean square deviation (RMSD) derived from MD trajectories for complexes of TLR3 with vaccine candidates of (**a**) NV6, (**b**) NV1, (**c**) NV4, and (**d**) NV5.

**Figure 7 vaccines-14-00050-f007:**
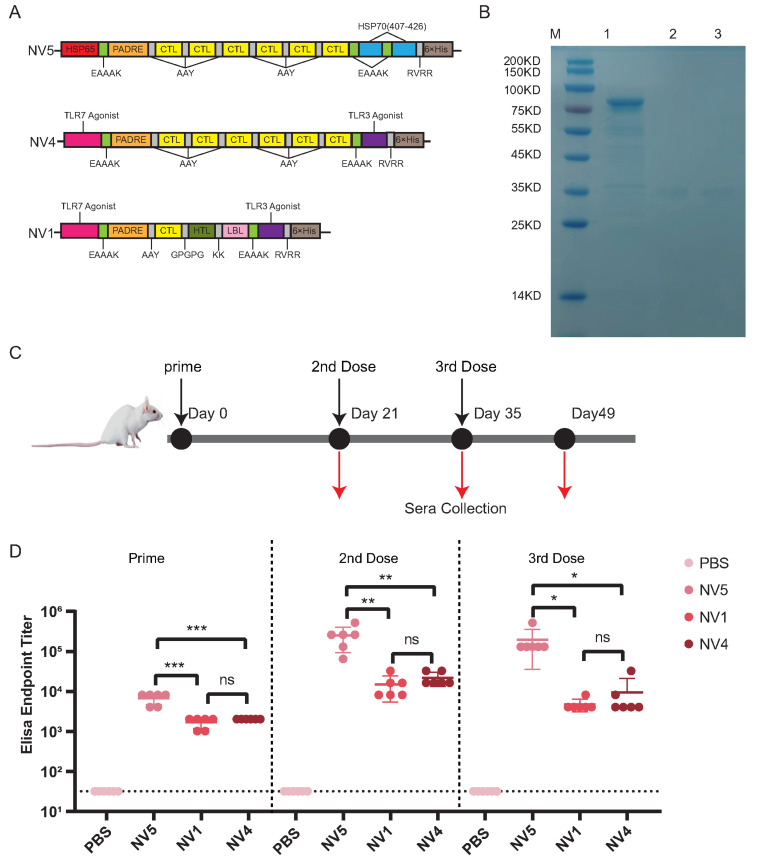
Characterization of vaccines NV5, NV4, and NV1. (**A**) Schematic constructs of the three vaccines. (**B**) SDS-PAGE analysis of purified proteins. Lane M, marker; lane 1, NV5 (76.9 kDa); lane 2, NV1 (27.3 kDa); lane 3, NV4 (30.8 kDa). (**C**) Immunization schedule: mice were subcutaneously immunized on days 0, 21, and 35, with sera collected on days 21, 35, and 49. (**D**) Antibody responses in mice immunized with PBS or the three vaccines. Complete Freund’s adjuvant was used for the first dose, incomplete Freund’s adjuvant for the boosters. ELISA endpoint titers were defined as the reciprocal of the highest serum dilution with absorbance ≥0.1 OD above blank. The antisera of the PBS group did not show any significant reactivity at 1:64 dilution (the lowest dilution tested) and were therefore assigned a titer of 32 for geometric mean titer (GMT) computation. Each symbol denotes one mouse; lines indicate group GMT. Each symbol denotes one mouse; lines indicate group GMT, data are mean values ± SD, n = 6. Statistical significance: ns, p>0.05; *, p<0.05; **, p<0.01; ***, p<0.001.

**Figure 8 vaccines-14-00050-f008:**
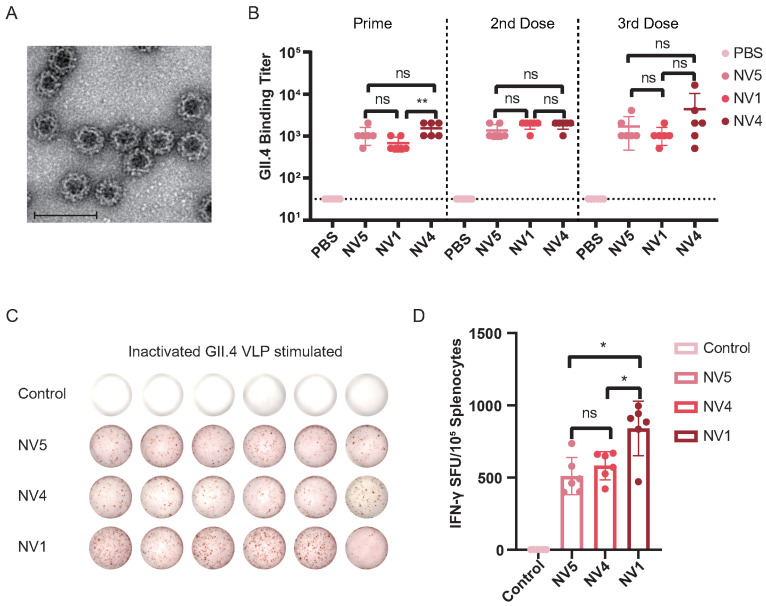
Immunogenicity of candidate norovirus vaccines NV5, NV4, and NV1 in mice. (**A**) Transmission electron microscopy of purified GII.4-VLP, negatively stained with 0.5% uranyl acetate. Scale bar = 100 nm. (**B**) GII.4-specific antibody titers in sera collected at days 21, 35, and 49 after the first immunization. Endpoint titers were defined as the reciprocal of the highest dilution with absorbance ≥0.1 OD above blank. PBS group sera showed no reactivity at 1:64 dilution and were assigned a titer of 32 for GMT calculation. Each symbol represents one mouse and the line indicates the geometric mean value of the group. (**C**,**D**) GII.4-specific IFN-γ responses measured by ELISpot. Representative well images and spot-forming units (SFU) from splenocytes after stimulation with inactivated GII.4-VLP are shown. Each symbol denotes one mouse; lines indicate group GMT, data are mean values ± SD, n = 6. Statistical significance: ns, p>0.05; *, p<0.05; **, p<0.01.

**Table 1 vaccines-14-00050-t001:** Properties of the CTL, HTL, and LBL epitopes selected to construct the vaccine candidates.

Epitope	Length	Alleles	Percentile Rank	Coverage	Hydropathicity
**CTL epitopes**					
TYPGEQILF	9	HLA-A*24:02/HLA-C*04:01	0.01	30.76%	0.01
NIIDPWIMK	9	HLA-A*11:01	0.04	43.48%	0.22
TLEPIFIPV	9	HLA-A*02:01	0.31	14.62%	1.38
LMAGNAFTA	9	HLA-A*02:01	0.32	14.62%	1.03
VLMAGNAFT	9	HLA-A*02:01	0.37	14.62%	1.30
**HTL epitopes**					
PVAGGAIAA	9	DQA1*05:01	0.03	32.43%	1.50
FTAGKVIFA	9	DQA1*05:01	0.05		1.43
**LBL epitopes**					
FFRSYIPLKGGFGNTA	16	-	-	-	0.36
LKGGFGNTAI	10	-	-	-	0.07

**Table 2 vaccines-14-00050-t002:** Predicted performance of the vaccine candidates selected to carry out structural prediction.

Vaccine Candidate	Antigenicity	Allergenicity	Solubility	Estimated Half-Life	Instability Index	Aliphatic Index	GRAVY
NV1 (7PCHB3)	0.5418	No	0.675	30 h Mammalian reticulated red in vitro >20 h Yeast in vivo >10 h *E. coli* in vivo	34.74	73.29	−0.402
NV2 (7PHBC3)	0.5170	No	0.675	30 h Mammalian reticulated red in vitro >20 h Yeast in vivo >10 h *E. coli* in vivo	35.13	73.29	−0.402
NV3 (7PBCH3)	0.5142	No	0.675	30 h Mammalian reticulated red in vitro >20 h Yeast in vivo >10 h *E. coli* in vivo	34.74	73.29	−0.402
NV4 (7PCCCCCC3)	0.5913	No	0.591	30 h Mammalian reticulated red in vitro >20 h Yeast in vivo >10 h *E. coli* in vivo	36.07	90.86	−0.064
NV5 (65PCCCCCCM2)	0.5103	No	0.703	30 h Mammalian reticulated red in vitro >20 h Yeast in vivo >10 h *E. coli* in vivo	31.20	103	−0.031
NV6 (7PC12345H12B123)	0.4391	No	0.640	30 h Mammalian reticulated red in vitro >20 h Yeast in vivo >10 h *E.coli* in vivo	31.20	75.55	−0.173

**Table 3 vaccines-14-00050-t003:** The interaction analysis of the docked complexes of TLR3 with NV6, NV1, NV4, and NV5.

	NV6-TLR3	NV1-TLR3	NV4-TLR3	NV5-TLR3
**Interface in ligand**	20.8%	22.2%	25.8%	10.7%
**Interface in receptor**	10.5%	10.3%	12.5%	15.2%
**Hydrogen bonds**	17	11	18	27
**Salt bridges**	6	17	17	11

## Data Availability

All the data generated in this study are included in the current manuscript and in the Appendix A and are not submitted to any database. Any specific data can be provided if needed.

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
