# Peer review of "Optimized Multi-Epitope Norovirus Vaccines Induce Robust Humoral and Cellular Responses in Mice"

_vaccines, 2025, doi:10.3390/vaccines14010050_

Round 1

Reviewer 1 Report

Comments and Suggestions for Authors

The approach for in silico design of multi-epitope vaccine candidates is a very widely used approach. One that has been previously applied to the design of norovirus vaccines. This manuscript described a very similar approach to those used in the past (eg. PMID: 40375772, PMID: 33170093, PMID: 31233780 and PMID: 40426240) with the addition of in vivo mouse immunizations which does add a notable addition to the previous reports

Points to be addressed:

- section 3.4 is a very interesting approach to utilise the C-IMMSIM software to predict any potential immune response. It is obviously very informative but it is just modelling. Throughout this section (both in the main text and Supplementary Figures) it is stated that this was the immune response generated but this is not true, it is the response that was modelled. Therefore, in all instances the text must be revised to reflect this eg. Line 442 “NV1 induced high antibody titres” should read “NV1 modelling revealed it could produce high antibody titres” or similar

- the most convincing results were those shown in Figs 7 and 8 where actual in vivo administration of the peptides was evaluated. However, in Fig 7b it would appear that NV1 and NV4 are migrating at a similar size (both around 35 kDa, not 31 and 27 kDa as suggested) which would be incorrect.

- Fig 7d it would suggest there was no boosting after the 2nd dose, was that correct? In fact antibody titres appeared to decline in all cases. Comment is required

- Fig 8, shows nice reactivity to GII.4 but no breadth in the antibody response. The premise of the report is to generate a more broadly reactive and neutralising response but only one genogroup was used. Subsequent VLPs need to be include to determine cross-reactivity. Additionally, at least some analyses needs to be included on whether the antibodies can prevent VLP interaction with HBGAs

- one of the known correlates of protection against norovirus infection is mucosal IgA, so was IgA induced upon immunization?

- Additionally, the ability to evaluate the addition of HLA epitopes was not included so the merit of their inclusions is still to be assessed and determined

Author Response

Comments 1:  section 3.4 is a very interesting approach to utilise the C-IMMSIM software to predict any potential immune response. It is obviously very informative but it is just modelling. Throughout this section (both in the main text and Supplementary Figures) it is stated that this was the immune response generated but this is not true, it is the response that was modelled. Therefore, in all instances the text must be revised to reflect this eg. Line 442 “NV1 induced high antibody titers” should read “NV1 modelling revealed it could produce high antibody titers” or similar.

Replies 1: Thank you for this insightful comment. We fully agree that the outputs of C-IMMSIM are modelled predictions rather than experimentally generated immune responses. We have revised the language throughout Section 3.4 to clearly indicate that C-IMMSIM outputs are modelled predictions, not experimentally generated responses. All corresponding changes have been highlighted in the revised manuscript (Lines 351-373 on Pages 11).

Comments 2: - the most convincing results were those shown in Figs 7 and 8 where actual in vivo administration of the peptides was evaluated. However, in Fig 7b it would appear that NV1 and NV4 are migrating at a similar size (both around 35 kDa, not 31 and 27 kDa as suggested) which would be incorrect..

Replies 2: Thank you for this insightful comment. We have revised the language throughout Section 3.8 and provided an explanation. These proteins are unusually basic(NV4:K/R/H ≈19.4% of residues,NV1:K/R/H ≈ 23% of residues)and rich proline ( NV4 : P ≈ 6.5% of residues , NV1 : P ≈ 4.8% of residues ) , these features can reduce uniform SDS binding and result in slower migration on SDS–PAGE, producing an apparently higher apparent molecular weight. All corresponding changes have been highlighted in the revised manuscript (Lines 447-453 on Pages 15).

Comments 3: Fig 7d it would suggest there was no boosting after the 2nd dose, was that correct? In fact antibody titers appeared to decline in all cases. Comment is required.

Replies 3: Thank you for this insightful comment. We confirm that the ELISA data indicate no further rise in antibody titer after the 2nd dose and, in fact, a decline in endpoint titers measured at the sampling timepoint after the 3rd dose. We have re-checked the raw ELISA data and included the GMT ± SD and formal statistical comparison (endpoint titers ± SD, n = 6) between post-second and post-third timepoints in the revised Supplementary Figure S11 (Figure S11 on Page 13 in the supplementary material). This result shows there was no boosting after the 2nd dose, this may be because the antibody response had already reached its peak after the second dose.

Comments 4: Fig 8, shows nice reactivity to GII.4 but no breadth in the antibody response. The premise of the report is to generate a more broadly reactive and neutralising response but only one genogroup was used. Subsequent VLPs need to be include to determine cross-reactivity. Additionally, at least some analyses needs to be included on whether the antibodies can prevent VLP interaction with HBGAs.

Replies 4: Thank you for this insightful comment. Our results showed that the vaccine can generate GII.4-specific antibodies but the vaccine can’t block VLP binding with its receptor HBGAs. As a prove of conscept study, our study only focus only one surface linear epitope from GII.4. So the resulting antibodies showed GII.4 VLP interaction only and barely prevent VLP binding with HBGAs. In future studies, we will focus on designing vaccines that incorporate multiple epitopes from different norovirus serotypes, so that we can generate broad neutralizing antibodies. Relevant content regarding this limitation and our future work has been added to the Discussion and Conclusion sections of the revised manuscript. All corresponding changes have been highlighted in the revised manuscript (Lines 533-536 on Pages 19).

Comments 5: one of the known correlates of protection against norovirus infection is mucosal IgA, so was IgA induced upon immunization?

Replies 5: Thank you for this great comment. Mucosal IgA is considered providing critical protection against norovirus infection, but how to generate IgA by vaccination is still a challenge for norovirus vaccine development . In this study, we focused primarily on GII.4 VLP specific antibody responses and cellular immune response. We also tested mucosal IgA, but it was barely induced by our strategy. We have acknowledged this limitation in the revised manuscript (Lines 551 on Pages 20) and will include evaluation of mucosal IgA responses in future studies.

Comments 6: Additionally, the ability to evaluate the addition of HLA epitopes was not included so the merit of their inclusions is still to be assessed and determined.

Replies 6: We sincerely appreciate your insightful comment. The primary merit of incorporating specific HLA genotype restrictions into our design pipeline was to maximize the population coverage of the vaccine candidates. Specifically, we targeted the most prevalent HLA alleles identified in public databases to predict and screen T-cell epitopes, ensuring the vaccine could theoretically elicit an immune response in a vast majority of the population. The efficacy of this strategy is quantitatively reflected in Table 1, which presents the high population coverage rates calculated via the IEDB Population Coverage analysis tool.

Reviewer 2 Report

Comments and Suggestions for Authors

Manuscript by Xing et al. presents a comprehensive reverse-vaccinology–to–in vivo validation pipeline for multi-epitope norovirus vaccines. The authors systematically evaluate four construction strategies, perform extensive immune-informatics screening, and experimentally validate three representative vaccine constructs (NV1, NV4, NV5) in mouse models. The integration of computational prediction, structural modeling, in addition to immunogenicity testing is a major strength. The experimental validation strongly enhances the credibility of the computational approach.

The manuscript is well-organized, and the methods are detailed. The study offers meaningful insights into how epitope arrangement, adjuvant selection, and repetitive architecture influence humoral vs. cellular immunity. Overall, the work is suitable for publication after revision.

Major comments:

  1. How the four construction strategies were rationally chosen is not clear.
  2. What specific limitations of existing multi-epitope norovirus vaccine designs the authors are overcoming.
  3. Why TLR3/TLR7 agonists were prioritized over other innate immune adjuvants.
  4. For statistical analysis only t-tests were used, but comparisons involve ≥3 groups, requiring ANOVA with appropriate post-hoc tests.
  5. Graphs uses SD or SEM is not clear.
  6. Authors claim NV5 produce- exceptionally high GMTs. However, the early VLP-specific responses favor NV4, and later titers across groups are similar. Please clarify
  7. NV1 shows strong IFN-γ responses (Fig 8d), but authors should discuss why TLR7/TLR3 agonists favor a stronger Th1 phenotype.
  8. Control group is not clear for TLR agonist and HSP adjuvents experiment.
  9. Many URLs embedded in text disrupt readability- move them to a separate “Resources” table or supplementary material.

Minor comments:

  1. Revise grammar and phrasing throughout the manuscript for conciseness.
  2. Make Fig 2 more visible, not easy to interpret.
  3. Figure legends should explicitly state sample sizes.
  4. Clarify the expression of NV6 was abandoned due to predicted transmembrane helices—please provide the TMHMM output in supplementary material.
  5. Provide molecular weights in kDa directly in the SDS-PAGE figure or caption.
Comments on the Quality of English Language

Requires rigorous revision

Author Response

Comments 1: How the four construction strategies were rationally chosen is not clear.

Replies 1: Thank you for your good suggestions. We have revised the manuscript to provide a clearer, more concise, and professional explanation of how the four construction strategies were rationally chosen. Specifically, we clarified the design rationale in Methods Section 2.3.1 and aligned the corresponding outcomes in Results Section 3.3. All revisions are highlighted in the revised manuscript for easy review (Lines 160-188 on Pages 5, Figure 2 on Page 6).

Comments 2: What specific limitations of existing multi-epitope norovirus vaccine designs the authors are overcoming.

Replies 2: Thank you for this important comment. The primary limitations of current multi-epitope norovirus vaccine designs that our study are overcoming are twofold:

  • Lack of Experimental Validation: Most previous efforts remained at the in silico design stage. Our study bridged this gap by providing in vivo validation via murine immunogenicity and cellular immunity assays.
  • Reliance on Single Construction Strategies: Previous studies used a single construction method without exploring how different constructs affect structure and immune efficacy. In contrast, this work systematically compared four construction strategies and interpreted the roles of epitope breadth, topological positioning, and adjuvant class in the relationships among sequence, structure, and immune function.

These improvements were added in Introduction (Lines 58-67 on Pages 2).

Comment 3: Why TLR3/TLR7 agonists were prioritized over other innate immune adjuvants.

Replies 3: Thank you for the question. We chose to prioritize TLR3 and TLR7 agonists as adjuvants in our sequence design for the following reasons:

  • Unlike TLR2 or TLR4 agonists, which primarily activate the NF-κB pathway to induce pro-inflammatory cytokines, TLR3 and TLR7 serve as endosomal sensors. Their activation triggers the IRF3 and IRF7 signaling pathway, leading to the substantial production of Type I Interferons (IFN-α/β), which are the critical cytokines for establishing an antiviral environment.
  • The activation of TLR3 and TLR7 in dendritic cells induces Type I IFNs and IL-12, enhances the expression of costimulatory molecules, and promotes a Th1 biased CD4+ response and CD8+ cytotoxic T cells (CTLs). This downstream signaling pattern is consistent with our goal to elicit a more strong immune response using the Norovirus multi-epitope vaccine.
  • Technically, TLR3/TLR7 agonist motifs can be conveniently constructed as short peptide sequences at the N- and C-termini of the fusion protein.

We have added a concise statement of this revision in Methods Section 2.3.1 and Discussion to facilitate readers understanding, with changes highlighted in the revised manuscript (Lines 162-164 on Pages 5, Lines 512-517 on Page 19).

Comments 4: For statistical analysis only t-tests were used, but comparisons involve ≥3 groups, requiring ANOVA with appropriate post-hoc tests.

Replies 4: Thank you for this helpful comment. We fully agree that statistical analysis used ANOVA with appropriate post-hoc tests. We have revised the statistical analysis in the revised manuscript (Lines 293-295 on Pages 8).

Comments 5: Graphs uses SD or SEM is not clear.

Replies 5: Thank you for this helpful comment. Data are mean values ± SD. We have revised the figure legend throughout fig7 and fig8. All changes are highlighted in the revised manuscript (Lines 293 on Pages 8, Figure 7 and 8 on Page 17-18). 

Comments 6: Authors claim NV5 produce- exceptionally high GMTs. However, the early VLP-specific responses favor NV4, and later titers across groups are similar. Please clarify.

Replies 6: Whank you for raising this point. We have added a clarification in the Discussion section (Lines 503–511, Page 19 of the revised manuscript). Specifically, while we observed that NV5 elicits exceptionally high GMTs—likely due to the strong immune stimulation by HSP65 and HSP70—the early VLP-specific responses were indeed stronger for NV4. This is because TLR7 and TLR3 in NV1 and NV4 promote faster VLP-specific responses, and the multiple tandem CTL epitopes in NV4 further accelerate the induction of specific antibodies. In later stages, titers become comparable across groups due to the shared TLR sequence present in all constructs.

Comments 7: NV1 shows strong IFN-γ responses (Fig 8d), but authors should discuss why TLR7/TLR3 agonists favor a stronger Th1 phenotype.

Replies 7: Thank you for your consideration. We have added an explanation in the Discussion. In brief, NV1 contains TLR7 and TLR3 agonists, both of which promote a strong Th1-biased response by inducing type I interferons, IL-12, and robust dendritic-cell activation. TLR7 stimulation in plasmacytoid DCs drives IFN-α/β production, whereas TLR3 activation triggers TRIF-dependent signaling and high IL-12 levels. These cytokines collectively enhance CD4⁺ Th1 differentiation and promote IFN-γ production, which is consistent with the strong IFN-γ responses observed in Fig. 8d. This has been explained in Lines 512-517 on Pages 19. 

Comments 8: Control group is not clear for TLR agonist and HSP adjuvants experiment.

Replies 8: Thank you for this suggestion regarding the presentation of the control data. We apologize for the lack of clarity regarding the control group in the TLR agonist and HSP adjuvants experiment.In the revised manuscript, we have included results from the detection of GII.4-specific antibodies in sera obtained from immunizations that contained only the TLR agonist and HSP adjuvants (these sera were sourced from a separate, unrelated study). The data confirm that the adjuvants alone do not induce GII.4-specific antibody responses. The corresponding ELISA raw data have been added to the “Raw Data” zip file.

Comments 9: Many URLs embedded in text disrupt readability- move them to a separate “Resources” table or supplementary material.

Replies 9: Thank you for this helpful suggestion on presentation. We have removed embedded URLs from the main text and consolidated all links to web servers and databases into a dedicated appendix titled “Web Servers and Databases” at the end of the manuscript (for easy reference). The main text now cites these resources by name only. All changes are highlighted in the revised manuscript (Lines 589-619 on Pages 20-21). 

Comments 10: Revise grammar and phrasing throughout the manuscript for conciseness.

Replies 10: We thank the reviewer for this suggestion. The manuscript has been carefully reviewed and edited to improve grammar, phrasing, and overall conciseness. The text has been thoroughly polished to ensure clarity and readability.

Comments 11: Make Fig 2 more visible, not easy to interpret.

Replies 11: We appreciate the reviewer’s feedback regarding the clarity of Figure 2. We have redesigned this figure with higher resolution and optimized formatting to ensure it is now more visible and easier to interpret (Figure 2 on Page 6). 

Comments 12: Figure legends should explicitly state sample sizes.

Replies 12: Thank you for this helpful comment. We have added sample sizes in figure legends. Please see the legends of Figures 7 and 8 on Page 17-18. 

Comments 13: Clarify the expression of NV6 was abandoned due to predicted transmembrane helices—please provide the TMHMM output in supplementary material.

Replies 13: Thank you for your good suggestions. We have clarified that NV6 expression was not pursued due to predicted transmembrane helices, and we have now added the TMHMM prediction outputs (per-residue probability) to the Supplementary Materials Figure S10 to substantiate this point. The main text has been updated to cite the new supplementary item, and all changes are highlighted in the revised manuscript. 

Comments 14: Provide molecular weights in kDa directly in the SDS-PAGE figure or caption.

Replies 14: Thank you for this helpful comment. We have added molecular weights in figure legends. Please see the legends of Figure 7 on Page 17.

Reviewer 3 Report

Comments and Suggestions for Authors

The study presented in the article is undoubtedly relevant because there is an urgent need in developing an effective vaccine towards norovirus, which is a prevalent cause of gastroenteritis worldwide and for which no licensed drugs or vaccines exist so far. Traditional approaches have failed to combat its rapid evolution and antigenic diversity. Authors successfully combine modern immunoinformatic methods and molecular modeling with experimental validation, allowing them to proceed from theoretical design to functional vaccine candidates.

A significant advantage of this work is the systematic approach to vaccine construction. The analysis covers various strategies for assembling epitopes and adjuvants, identifying key principles for directed formation of humoral and cellular immunity. The introduction contains all necessary references for understanding the significance of both theoretical and experimental work. Experiments are logically performed and described well enough, demonstrating a clear step-by-step transition from calculative prediction and screening of epitopes through designing and modeling vaccines to obtaining them and studying their immunological properties on animal models. The reliability of data obtained is validated by using standardized biochemical techniques (expression, purification, SDS-PAGE) and reproducible immunological tests (ELISA, ELISPOT), confirming predictions during immunoinformatic analysis. Figures and tables provide informative insights into the results. Conclusions are supported by results due to comprehensive validation that includes molecular docking, dynamic stability modeling of constructs, and cross-reactivity testing of induced antibodies with authentic virus-like particles of norovirus. This makes the paper interesting for a wide audience. However, several improvements could be considered, so, The manuscript can be published after minor revision.

  • Chapter 4 seems to be designed not so perfect. A part of information (lines 559-569) can be moved to Introduction part, and the rest of phrases (about limitations and future directions of research) – into the Conclusion one.
  • After moving some phrases from Section 4 to other sections, it would be beneficial to complement this part of the article with schematic images illustrating final vaccine candidates (NV1, NV4, NV5) clearly marking epitopes, linkers, and adjuvants. Highlighting these details in the main text will help readers immediately understand their fundamental differences, since currently one has to frequently refer back to the Methods section to get this information.
  • In the Introduction and Discussion, the addressed problem should be outlined more explicitly, in particular, emphasizing the lack of experimental verification and comparison of different strategies in previous studies
  • Some data can be moved from the overdetailed Methods section into supplementary materials, for example, complete lists of algorithms used for predicting epitopes.
  • It would be useful to explain why candidate NV5, which demonstrated outstanding antibody titers, elicited relatively weak cell-mediated responses compared to NV1.
  • Clearer parallels between the results of molecular docking or stability of complex with TLR3 and in vivo immune response strength should be provided.

Author Response

Comments 1: Chapter 4 seems to be designed not so perfect. A part of information (lines 559-569) can be moved to Introduction part, and the rest of phrases (about limitations and future directions of research) – into the Conclusion one..

Replies 1: We sincerely appreciate this constructive suggestion regarding the manuscript's structure. Accordingly, we have moved the information previously located in lines 559-569 to the Introduction section (Lines 58-67 on Pages 2). Furthermore, we have relocated the discussion on future directions to the Conclusion section, which now also includes a concise summary of the study's limitations (Lines 566-568 on Pages 20)   

Comments 2: After moving some phrases from Section 4 to other sections, it would be beneficial to complement this part of the article with schematic images illustrating final vaccine candidates (NV1, NV4, NV5) clearly marking epitopes, linkers, and adjuvants. Highlighting these details in the main text will help readers immediately understand their fundamental differences, since currently one has to frequently refer back to the Methods section to get this information..

Replies 2: Thank you for this helpful suggestion. We now display schematic representations of the final candidates (NV1, NV4, NV5) in Fig.7a, with epitopes, linkers, and adjuvants clearly annotated and color-coded. We also added brief descriptive sentences in the Discussion to highlight their fundamental differences, and included figure citations to direct readers to the corresponding structural models. The changes are highlighted in the revised manuscript (Lines 539 on Pages 19).

Comments 3: In the Introduction and Discussion, the addressed problem should be outlined more explicitly, in particular, emphasizing the lack of experimental verification and comparison of different strategies in previous studies.

Replies 3: We sincerely appreciate this valuable suggestion. Following your guidance, we have integrated a more explicit outline of the addressed problem into the Introduction section (Lines 58-67 on Pages 2). Specifically, we have emphasized the limitations of previous studies—namely, the lack of experimental verification and the absence of systematic comparisons between different construction strategies—to provide a clearer and more detailed context for our work.

Comments 4: Some data can be moved from the overdetailed Methods section into supplementary materials, for example, complete lists of algorithms used for predicting epitopes..

Replies 4: We appreciate the reviewer’s suggestion to improve the conciseness of the manuscript. Accordingly, we have significantly streamlined the Methods section by removing overdetailed descriptions. For the specific algorithms and underlying principles used for epitope prediction, we have focused on the core methodology and directed readers to the original cited references for comprehensive details.

Comments 5: It would be useful to explain why candidate NV5, which demonstrated outstanding antibody titers, elicited relatively weak cell-mediated responses compared to NV1.

Replies 5: We thank the reviewer for this observation. NV5 includes HSP65 and HSP70, which primarily function as potent B-cell and antigen-presenting cell chaperones, enhancing antibody production and humoral immunity. In contrast, NV1 contains TLR7 and TLR3 agonists, which strongly stimulate dendritic cells and promote Th1 polarization, leading to robust cell-mediated responses and IFN-γ production. Therefore, while NV5 elicits outstanding antibody titers, its adjuvant composition is less efficient at inducing Th1-biased cellular immunity, explaining the relatively weaker cell-mediated responses compared to NV1. We have added this clarification in the revised Discussion (Lines 503-517 on Pages 19).

Comments 6: Clearer parallels between the results of molecular docking or stability of complex with TLR3 and in vivo immune response strength should be provided.

Replies 6: We thank the reviewer for this observation. In the revised Discussion section, we have explicitly provided clearer parallels between the molecular docking results (specifically the interaction analysis and binding energy) and the strength of the in vivo immune response. We highlighted that the superior structural stability of the NV1-TLR3 complex correlates with the robust cellular immunity observed in the animal experiments, thereby establishing a logical connection between the in silico predictions and biological outcomes. The changes are highlighted in the revised manuscript (Lines 517-520 on Pages 19).

Round 2

Reviewer 1 Report

Comments and Suggestions for Authors

The authors have addressed a majority of my concerns but I still have 2 points that need some comment. 

  • Firstly what precedent to have to your reply to my comment 3 "This result shows there was no boosting after the 2nd dose, this may be because the antibody response had already reached its peak after the second dose". Are the reports that have shown this because the whole point of a booster is to increase the specificity and amount of antibody. Why would this not happen with your antigen?
  • Secondly, I don't believe the title of the paper duly reflects the results. As indicated by the authors they didn't generate broad or necessarily robust humoral responses in mice

Author Response

Comments 1: Firstly what precedent to have to your reply to my comment 3 "This result shows there was no boosting after the 2nd dose, this may be because the antibody response had already reached its peak after the second dose". Are the reports that have shown this because the whole point of a booster is to increase the specificity and amount of antibody. Why would this not happen with your antigen?

Replies 1: We agree that the fundamental purpose of a booster is to enhance antibody levels and affinity. Our observation that the third dose did not significantly increase the geometric mean titer (GMT) compared to the second dose requires careful explanation, and we believe it is attributable to a combination of methodological and biological factors.

First, a critical technical point: The serum samples collected after the second immunization and those collected after the third immunization were not assayed at the same time. This temporal separation in the ELISA runs can introduce well-known inter-assay variability, which may account for the slight numerical differences observed in the titers. Therefore, a direct, absolute comparison of the raw titers between these two time points should be made with caution.

However, despite this assay variability, the key finding is that the difference did not reach statistical significance. When we account for this variability through appropriate statistical testing, we find no evidence that the third dose elicited a further increase in antibody titer.

Biologically, this statistically non-significant result aligns with the interpretation that the humoral response may have reached a plateau after the second immunization. It is documented in this paper about Norovirus VLP vaccine phase I clinical study(Treanor JJ, et al., J Infect Dis. 2014). The high dose group (150ug) serum antibody GMT at day28 was high than that at day56 (third dose) both in GI.1 and GII.4 group. This happens in immunology when repeated immunization with the same antigen can lead to a saturation of the responding B cell clones, after which additional boosts primarily sustain rather than elevate the peak antibody level. We posit that our vaccine candidate, under the given formulation and schedule, elicited a strong enough primary response that approached its peak by the second dose.

Comments 2: Secondly, I don't believe the title of the paper duly reflects the results. As indicated by the authors they didn't generate broad or necessarily robust humoral responses in mice

Replies 2: Thank you for your suggestion regarding the paper title. We appreciate the opportunity to clarify our intent. Our original title, “Optimized Multi-Epitope Norovirus Vaccines Induce Robust Humoral and Cellular Responses in Mice”, aimed to highlight the multi-epitope design and the induction of both humoral (with titers up to 10^6) and cellular responses. We did not intend to claim “broad” neutralization in the title, as we explicitly state in the manuscript that breadth is limited by the inclusion of GII.4 epitopes only. We appreciate your point that the results demonstrate potent but strain-specific immunity, and we agree that the title should accurately reflect this. Upon careful consideration, we believe our original title remains accurate and appropriate.

Reviewer 2 Report

Comments and Suggestions for Authors

Thank you for addressing my concerns, now it looks more clear.

Comments on the Quality of English Language

Improved

Author Response

Comments from reviewer: Thank you for addressing my concerns, now it looks more clear.

Reply and revisions: We greatly appreciate your positive feedback. We are pleased to learn that our previous revisions have successfully addressed your concerns and improved the clarity of the manuscript.